# Visual Tokens Are Not Equal: Alleviating Hallucination in Multimodal Large Language Models via Aligning Attention

## Abstract

Hallucination remains a significant challenge for Multimodal Large Language Models (MLLMs), hindering their reliability across various tasks. Despite extensive research from various perspectives, the underlying causes remain unclear. In this paper, we conduct empirical analyses and identify a progressive attention shift in the decoding process, where the decoder's attention over visual tokens gradually diverges from the vision encoder's. Based on these observations, we infer that this shift systematically reduces the model's focus on semantically important visual tokens, leading to hallucinations. Building on this finding, we propose Align Attention with Image (AAI), a decoding-time method that explicitly aligns the decoder's attention over visual tokens with the self-attention of the vision encoder. Specifically, AAI caches the encoder's visual self-attention and leverages it as a reference signal to guide the decoder's attention distribution toward that of the image. AAI is decoding-agnostic and can be seamlessly integrated with both classical and modern decoding strategies across different MLLMs. We evaluate AAI on widely used hallucination benchmarks and show that it consistently reduces hallucinations without sacrificing semantic completeness. All relevant experimental code is included in the supplementary appendix and will be released publicly.

## 1 Introduction

Recent advances in Multimodal Large Language Models (MLLMs) have led to remarkable breakthroughs and broad applications, particularly in Vision-Language Models (VLMs). Therefore, this paper primarily focuses on VLMs; however, analysis and the proposed method can be readily extended to accommodate more modalities. Despite MLLMs' successes (Yang et al., 2025; Achiam et al., 2023; Chen et al., 2025b), they still continue to face a critical challenge: hallucination. (Jiang et al., 2025; Liu et al., 2024b). Specifically, models often frequently fabricate nonexistent objects (Campbell et al., 2025), miss existing ones (Wang et al., 2024b), or misattribute visual properties (Chen et al., 2025a). This issue significantly constrains the deployment of MLLMs in safety-critical domains, such as autonomous driving (Li et al., 2025) and medical AI (Pan et al., 2025).

Hallucinations in MLLMs stem from a range of complex factors. While some prior work has attributed hallucinations in Uni-Modal LLMs primarily to the influence of learned textual priors (Huang et al., 2025), other studies have argued that modality imbalance is the key factor in MLLMs, inspiring efforts to suppress textual dominance or to strengthen visual grounding (Zou et al., 2025; Liu et al., 2025). Despite recent advances, existing methods do not fundamentally resolve hallucination. Simply amplifying visual signals fails to induce genuine image re-examination and often results in overly conservative outputs that omit real objects. To this end, we further investigate another underexplored factor, **intra-modal** attention shifting, and its contribution to hallucinated generation.

In this paper, we conduct a comprehensive investigation into the underlying causes of hallucinations in MLLMs and identify a phenomenon we refer to as "attention misalignment." Our experiments

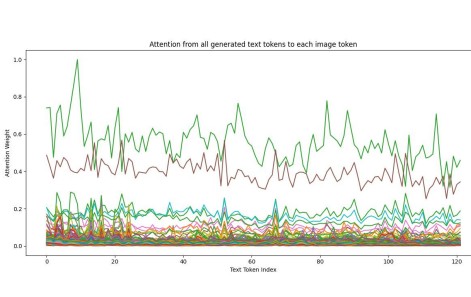

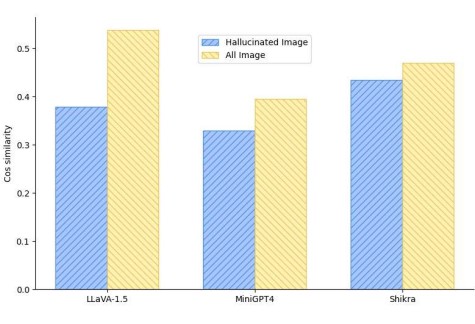

(a) Attention to Different Image Token

(b) Cosine Similarity of Attention

Figure 1: (a) Variation in attention weights assigned to different image tokens; (b) shows the cosine similarity between the vision encoder's self-attention and the decoder's attention over image tokens, comparing hallucinated images with all images.

reveal that this misalignment arises during the decoding process of MLLMs and plays a significant role in the emergence of hallucinations.

**Visual tokens Are Not Equal.** As shown in Fig. 1(a), MLLMs allocate attention substantially unevenly across image tokens, a pattern consistent with prior work. Prior studies (Fazli et al., 2025) suggest that balancing attention across image tokens may alleviate hallucinations in MLLMs. However, we argue that such imbalanced attention is intuitive, as humans naturally focus more on salient regions within an image. Our analysis reveals that such imbalance is not the root cause of hallucinations. Instead, hallucinations occur when, during decoding, the model's attention over image tokens shifts away from the encoder's self-attention, which typically captures salient content. (Details in Sec 3).

**Align attention by dynamically reweighting to correct discrepancy.** Building on this finding, we propose Align Attention with Image(AAI), a method designed to mitigate hallucination in MLLMs. The core idea is to cache the self-attention of the image encoder and use it as a reference during decoding to guide the model's attention distribution toward the visual content. Across existing hallucination evaluation benchmarks and metrics, AAI demonstrates a strong capability to effectively reduce hallucinations. Furthermore, GPT-4o–assisted evaluations and strong results on MME confirm the effectiveness of our method across diverse MLLM architectures, highlighting its broad capability.

**Contribution.**The main contribution of our paper can be summarized as follows:

- Our AAI method alleviates hallucinations in MLLMs without requiring any additional training and can be seamlessly integrated with various decoding strategies and other hallucination mitigation techniques.

- Our in-depth preliminary experiments and analysis reveal one of the underlying causes of hallucinations in MLLMs: during decoding, attention over image tokens shifts away from the encoder's self-attention, leading the model to neglect critical tokens.

- We conducted extensive experiments, including hallucination evaluation, general capability evaluation, and GPT-4o-Assisted evaluation. The results demonstrate that AAI effectively reduces hallucinations while preserving overall generation quality.

## 2 RELATED WORK

**MLLMs and VLMs.** In recent years, Large Language Models (LLMs) have rapidly advanced, driven by increasing computational resources. Notable examples include GPT (Brown et al., 2020), LLaMA (Touvron et al., 2023), and DeepSeek (Liu et al., 2024a), which have demonstrated substantial gains in text understanding and generation. Building on this success, Multimodal Large

Language Models (MLLMs) have made significant progress. MLLMs leverage models such as CLIP (Radford et al., 2021) and BLIP (Li et al., 2022) to project data from different modalities into a shared textual space, providing rich multimodal context to LLMs. Among them, vision-language models (VLMs) such as LLaVA (Liu et al., 2023), InstructBLIP (Dai et al., 2023), GPT-4V (Achiam et al., 2023), and DeepSeek-VL (Lu et al., 2024) have achieved notable success in multimodal understanding and generation.

**Mitigation of MLLMs Hallucination.** Hallucination in multimodal large language models (MLLMs) has motivated a range of decoding-time and verification-based strategies. A major line of research focuses on guided or constrained decoding, where token probabilities are reweighted to enforce stronger visual grounding. This includes CLIP-guided signals(CGD) (Deng et al., 2024), image-biased or image-dependent constraints(IBD and M3ID) (Zhu et al., 2025; Favero et al., 2024), and adaptive mechanisms such as focal-contrast(HALC) (Chen et al., 2024), global-local attention(AGLA) (An et al., 2025), residual visual cues(RVD) (Zhong et al., 2024), and dynamic correction(DECO) (Wang et al., 2024a). Others leverage high-level semantic or summarization guidance(MARINE and SUMGD) (Zhao et al., 2024; Min et al., 2025). Complementarily, retrospective approaches verify or refine generation after initial decoding, such as resampling-based verification(REVERSE) (Wu et al., 2025) and memory-space visual retracing(MemVR) (Zou et al., 2025). In contrast, beam-search-specific methods like OPERA (Huang et al., 2024) penalize overtrust in language priors by comparing beam candidates. While these methods alleviate hallucinations to some extent, they do not fundamentally resolve the problem, as they overlook shifts in intra-modal attention distributions. In this work, we introduce a new perspective on mitigating hallucinations in MLLMs and, building on this insight, propose an effective solution.

# 3 ANALYSIS

In this section, we conduct a series of empirical experiments and analyses to explore the underlying mechanisms of hallucination in MLLMs. To balance complexity and reliability, we evaluate LLaVA-1.5, MiniGPT-4, and Shikra. Unless otherwise noted, experiments use randomly sampled COCO subsets, following prior workWang et al. (2024a).

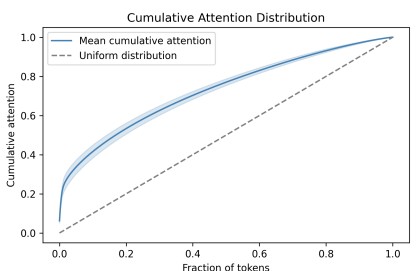

Figure 2: Cumulative Attention.

## 3.1 VISUAL TOKENS ARE NOT EQUAL.

Inspired by prior work, we randomly sampled 500 images for Different MLLMs to compute the attention distribution over visual tokens during decoding. In Fig 2, the blue curve denotes the mean cumulative attention distribution (shaded $\pm 1\sigma$), while the gray curve represents a uniform distribution. As shown, attention is heavily concentrated on a few "elite tokens" that dominate generation. Consequently, if the model attends to incorrect elite tokens, unintended hallucinations may arise.

## 3.2 ATTENTION MISALIGNMENT IN HALLUCINATION GENERATION.

To examine this phenomenon more clearly, we visualized attention for each image by projecting it into three dimensions with PCA and rendering it as RGB maps. We consistently observed a distinct "Attention Misalignment."

For a running example, as illustrated in Fig 3, both hallucinated and non-hallucinated cases exhibit structural patterns and salient object-focused attention in the Vision Encoder. However, during decoding of hallucinated images, attention diverges from the encoder patterns, showing a systematic shift across the entire process, not confined to the specific hallucination token. In contrast, non-hallucinated images preserve attention distributions consistent with the encoder for both semantically rich entity tokens and weaker abstract tokens. We therefore infer that when attention misalignment occur during decoding and underestimates elite tokens, hallucinations are likely to

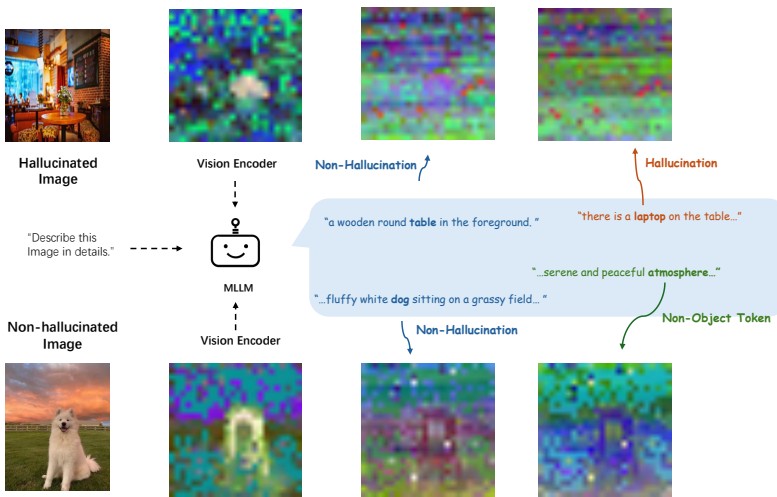

Figure 3: Attention distributions of hallucinated and non-hallucinated images (visualized as RGB maps via PCA). Shown are the encoder, hallucinated images with hallucination and non-hallucination tokens, and non-hallucinated images with object and abstract tokens.

emerge. Fig 1(b) confirms this, showing that hallucinated cases exhibit markedly lower cosine similarity than the overall average across all tested images, which indicates a significant positive correlation between hallucination occurrence and attention shift.

## 4 METHOD

Our proposed Align Attention with Image (AAI) approach is motivated by the key finding: during the decoding process, the attention distribution over image tokens progressively diverges from the self-attention distribution within the image encoder, and this misalignment is identified as one of the primary causes of hallucination in MLLMs. Intuitively, while the encoder focuses on semantically important regions, the decoder's attention drifts as textual context accumulates, leading to the neglect of crucial visual information and the generation of hallucinated content.

Building on this observation, our proposed method caches the self-attention distributions from the image encoder and use them to guide the decoder's attention towards better alignment with the original visual saliency. This strategy encourages the attention distribution over image tokens to better reflect the true importance of visual information during decoding. Additionally, we further re-weight the contributions of image and text modalities during decoding.

### 4.1 CACHE THE ATTENTION OF ENCODER

Our preliminary experiments, consistent with prior study (Wei et al., 2023) show that self-attention in vision encoders such as CLIP tends to focus on salient tokens or semantically rich regions. This observation suggests that the distribution of self-attention inherently captures the importance of different visual tokens. Motivated by this, we cache the self-attention of every head and every layer in the Vision Encoder of the MLLM, denoted as $A^E \in \mathbb{R}^{L \times H \times T \times T}$, where $L$ denotes the number of encoder layers, $H$ denotes the number of heads in self-attention and $T$ denotes the sequence length of image tokens. This attention matrix quantifies the degree to which each image token attends to other image tokens. By aggregating the attention scores along the token dimension, we obtain a measure of how much each token is attended to by the model, as defined in Eq. 1.These cached attention maps will serve as reference distributions, encouraging alignment with the regions.

$$A_{ref}^E = \frac{1}{H} \sum_{h=1}^{H} \sum_{j} A_{L,h,:,j}^E \qquad (1)$$

Figure 4: Overview of the proposed method.

where $A_{ref}^{E} \in \mathbb{R}^{1 \times T}$ denotes the self-attention matrix over $T$ image tokens, corresponding to summing along the token dimension. Empirically, we select final layer and average over all attention heads as our reference cache, since the final layer captures sufficient information while individual heads encode diverse aspects.

For certain MLLMs that employ a BERT-style Q-Former to compress visual tokens, we additionally cache the Q-Former's attention and multiply it with the image self-attention as a substitute reference distribution.

## 4.2 ALIGN ATTENTION WITH IMAGE

Several studies have demonstrated that one of the primary causes of hallucination in MLLMs is the imbalance between the visual and textual modalities. Prior works mainly address this by re-injecting visual information (Zou et al., 2025) or amplifying the visual modality's weight (Liu et al., 2025). However, such straightforward methods often cause the model to overlook important image details, leading to object omissions. As analyzed in Section 1, another critical underexplored factor contributing to hallucination is the misalignment within the self-attention distribution among image tokens. To address this, we propose Align Attention with Image (AAI), which explicitly corrects attention shifts among image tokens to better align the model's focus with the actual visual content.

As shown in Fig. 4, we use the mean-normalized attention cache $A_{ref}^{E}$ from the vision encoder as a reference mask to guide the decoder's attention towards the visual modality. This encourages the decoder to focus on tokens emphasized by the vision encoder, promoting content that better reflects the image. We introduce the hyper-parameter $\tau$ to control the strength of the mask guidance, employing the absolute values of the original attention weights to implement a "soft" rather than "hard" guidance strategy. This design allows us to maximize the preservation of the model's generation performance while effectively steering the attention distribution. The process is formally defined as follows:

$$Mask_{ref} = \frac{A_{ref}^{E}}{mean(A_{ref}^{E})} \tag{2}$$

$$A'_{k,j} = \underbrace{(A_{k,j} + \tau \cdot Mask_{ref} \cdot |A_{k,j}|)}_{\text{AAI term}} \cdot \underbrace{\frac{\sum_{j} A_{k,j}}{\sum_{j} (A_{k,j} + \tau \cdot Mask_{ref} \cdot |A_{k,j}|)}}_{\text{scaling factor}}, \quad \text{for } j = 1, \ldots, n_V \tag{3}$$

Eq 2 illustrates the construction of a soft guidance mask $Mask_{ref}$, by normalizing attention from the encoder cache, denoted as $A^E_{ref}$. $Mask_{ref}$ denotes the reference distribution during decoding, and AAI encourages the original attention distribution to align with it, thereby directing the MLLM to focus more on critical image tokens. Eq 3 defines the adjusted attention weight $A'_{k,j}$ for the last token. The AAI term applies soft guidance by multiplying the absolute $A_{k,j}$, attention of the $k$-th generated token to the $j$-th image token, with the reference mask $Mask_{ref}$ and a strength parameter $\tau$, followed by adding the residual of $A_{k,j}$. This residual connection preserves information during decoding, thereby enabling more stable generation. The second term, a rescaling factor, restores $A_{k,j}$ to its original magnitude to ensure semantic continuity.

### 4.3 WARM UP AND RE-WEIGHT ATTENTION

In Figure 5, it can be observed that hallucinations rarely occur in the early stages of token generation. We refer to this period as the "warm-up phase," during which the model tends to generate faithful content, relying more on retained visual signals and being less influenced by linguistic biases. To leverage this observation, we define a hyperparameter $t$ such that the AAI mechanism is activated starting from the $(t+1)^{\text{th}}$ token. This allows the model to maintain natural and coherent text generation during the initial phase while enabling targeted attention adjustment afterward.

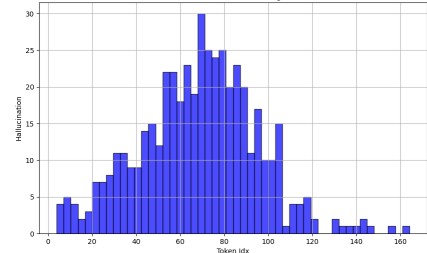

Figure 5: Hallucination frequency across 500 images in LLaVA-1.5.

To further mitigate hallucinations, we re-weight attention after applying AAI alignment to enhance the contribution of visual information. We introduce a hyperparameter $\beta$ to control the strength of this adjustment, formally defined as:

$$A''_{k,j} = \beta \cdot \underbrace{(A_{k,j} + \tau \cdot Mask_{ref} \cdot |A_{k,j}|)}_{\text{AAI term}} \cdot \underbrace{C}_{\text{scaling factor}}$$
$$= \beta \cdot A_{k,j} + \alpha \cdot Mask_{ref} \cdot |A_{k,j}|, \quad \text{where} \quad \alpha = \beta \cdot \tau \tag{4}$$

As $C$ serves as a constant normalization factor within each decoding step, it can be absorbed into $\beta$ and $\alpha$ without loss of generality, i.e., $\beta \leftarrow \beta C$ and $\alpha \leftarrow \alpha C$. Ultimately, $\alpha$ regulates the strength of attention alignment, while $\beta$ controls the relative attention allocated to image tokens within the overall token.

This approach of modulating attention to balance modalities is consistent with many prior studies (Li et al., 2024; Zhang et al., 2024; Liu et al., 2025). Essentially, this operation serves as a prompt for the model to refocus on visual information that may have been gradually forgotten during the generation.

## 5 EXPERIMENT

### 5.1 SETUP

**Baselines.** We integrated AAI into various decoding strategies, including greedy search, beam search, and nucleus sampling. To comprehensively evaluate AAI, beyond the standard decoding approach, we also consider representative hallucination-mitigation methods from different perspectives as baselines. These include MemVR (Zou et al., 2025), which re-injects visual information during decoding as supplementary evidence; HALC (Chen et al., 2024), which suppresses irrelevant objects through focal contrastive generation; CGD (Deng et al., 2024), which leverages CLIP-based relevance evaluation; PAI (Liu et al., 2025), which re-weights image contributions during decoding; VCD (Leng et al., 2024), which constrains token generation via contrastive decoding; and OPERA (Huang et al., 2024), which penalizes over-trust in early layers through beam-search–based decoding. All parameters for these methods are same to their original publications to ensure fair comparisons.

**Model.** We selected three popular vision-language models for evaluation: LLaVA-1.5 (Liu et al., 2023), MiniGPT-4 (Zhu et al., 2024), and Shikra (Chen et al., 2023), all in their 7-billion-parameter (7B) versions.

**Details.** To achieve better performance, we conducted ablation studies, as detailed in following section. The warm-up step was set to 5, and AAI was applied from layer 2 to last layer. For LLaVA-1.5 and MiniGPT-4, $\alpha$ was set to 0.4; for Shikra, $\alpha$ was set to 0.3, while $\beta$ was fixed at 1.0 for all models. In beam search, the beam size was fixed at 5, with all other settings following the original paper. To better observe hallucinations, the maximum generation length was set to 512.

## 5.2 BENCHMARK AND METRIC

**CHAIR. (Rohrbach et al., 2018)** The Caption Hallucination Assessment with Image Relevance (CHAIR) is a widely adopted benchmark for evaluating hallucinations in image captioning tasks. Specifically, CHAIR quantifies hallucination by measuring the proportion of objects mentioned in generated captions but absent from ground-truth annotations. It reports two complementary metrics: $CHAIR_S$ at the sentence level and $CHAIR_I$ at the instance level, defined as:

$$CHAIR_I = \frac{|\{hallucinated\ objects\}|}{all\ mentioned\ objects}, \quad CHAIR_S = \frac{|\{captions\ with\ hallucinated\ objects\}|}{all\ captions} \quad (5)$$

We conducted evaluations on the MSCOCO 2014 dataset. Following the experiment of OPERA (Huang et al., 2024), we used the prompt *"Please describe this image in detail."* to generate image captions. A subset of 500 images was sampled from the validation set for evaluation.

**POPE. (Li et al., 2023)** The Polling-based Object Probing Evaluation (POPE) is designed to assess hallucinations in Visual Question Answering (VQA) tasks. It formulates binary questions such as *"Is there a `<object>` in the image?"*, where `<object>` is substituted with items from three distinct splits: random, which includes objects randomly sampled from the dataset; popular, consisting of the most frequently occurring objects; and adversarial, which comprises objects that are highly correlated with the actual objects present in the dataset. Similarly, we conducted experiments on 500 images from the COCO dataset, with six questions evaluated for each split. F1 score was employed as evaluation metrics.

**MME. (Fu et al., 2024)** The MLLM Evaluation benchmark (MME) provides a comprehensive assessment of the perceptual and cognitive capabilities of MLLMs across 14 sub-tasks, including OCR, visual knowledge, and object recognition.

**GPT-4o assisted evaluation.** To further assess model performance and hallucination in image-grounded tasks beyond object hallucination captured by CHAIR, we extended our evaluation with an open-ended protocol. Following prior works (Huang et al., 2024) (Wang et al., 2024a), we randomly sampled 100 images from the COCO dataset and conducted an open evaluation using GPT-4o. We constructed prompts that, along with the image and two assistant-generated descriptions, were presented to GPT-4o for evaluation. GPT-4o assessed that along three dimensions: Accuracy (A), Detailedness (D) and Coherence (C) The full prompt design is shown in the appendix.

## 5.3 EXPERIMENT RESULTS

**Results on hallucination in image captioning.** Table 1 presents our image captioning evaluation results, demonstrating the effectiveness of AAI across three MLLMs, LLaVA-1.5 (Liu et al., 2023), MiniGPT-4 (Zhu et al., 2024), and Shikra (Chen et al., 2023), under greedy search, beam search, and nucleus sampling. It can be observed that AAI consistently outperforms competing approaches, achieving significant improvements over existing methods. Notably, AAI also demonstrates strong robustness, maintaining consistently superior performance across different models and decoding strategies.

Besides, while methods such as OPERA effectively reduce hallucinations, they incur substantial computational overhead. In contrast, AAI not only mitigates hallucinations, but also maintains inference efficiency comparable to vanilla decoding methods.

Table 1: CHAIR hallucination evaluation results. A lower score indicates reduced hallucination. Best results are highlighted in **bold**.

| Decoding | Method | LLaVA-1.5 | | MiniGPT-4 | | Shikra | |
|---|---|---|---|---|---|---|---|
| | | $CHAIR_S$ | $CHAIR_I$ | $CHAIR_S$ | $CHAIR_I$ | $CHAIR_S$ | $CHAIR_I$ |
| Greedy | Vanilla | 50.0 | 15.4 | 31.6 | 9.8 | 55.8 | 15.4 |
| | MemVR | 46.6 | 13.0 | 30.8 | 9.5 | 46.3 | 14.9 |
| | CGD | 45.7 | 12.7 | 30.8 | 9.2 | 45.1 | 13.3 |
| | PAI | 40.1 | 11.8 | 25.4 | 7.7 | 42.3 | 12.5 |
| | HALC | 38.8 | 9.7 | 28.1 | 9.1 | 40.5 | 12.1 |
| | AAI(Ours) | **37.4** | **9.2** | **22.7** | **7.2** | **39.8** | **11.7** |
| Beam Search | Vanilla | 48.8 | 13.8 | 30.2 | 9.4 | 50.2 | 13.3 |
| | OPERA | 44.6 | 12.8 | 25.8 | 9.4 | 36.0 | 12.0 |
| | AAI(Ours) | **41.0** | **11.6** | **25.4** | **8.5** | **37.1** | **11.7** |
| Nucleus | Vanilla | 49.6 | 14.9 | 32.7 | 11.9 | 55.6 | 15.4 |
| | VCD | 48.6 | 14.9 | 34.8 | 11.5 | 51.3 | 15.1 |
| | AAI(Ours) | **44.0** | **13.3** | **29.7** | **9.6** | **44.2** | **13.6** |

Table 2: POPE Hallucination Evaluation Results with Average F1 Scores across Random, Popular, and Adversarial Settings. Best results are highlighted in **bold**.

| Decoding | Method | LLaVA-1.5 $F1$ | MiniGPT-4 $F1$ | Shikra $F1$ |
|---|---|---|---|---|
| Greedy | Vanilla | 84.7 | 73.7 | 82.0 |
| | MemVR | 87.3 | 76.6 | 79.3 |
| | CGD | 86.7 | 77.4 | 80.1 |
| | PAI | 85.9 | 76.2 | 79.1 |
| | HALC | 87.2 | 75.9 | 82.6 |
| | AAI(Ours) | **89.2** | **83.1** | **83.1** |
| Beam Search | Vanilla | 84.9 | 70.3 | 82.5 |
| | OPERA | 85.4 | 73.3 | 82.7 |
| | AAI(Ours) | **89.7** | **79.9** | **83.5** |
| Nucleus | Vanilla | 83.1 | 58.5 | 81.2 |
| | VCD | 83.1 | 73.3 | 81.9 |
| | AAI(Ours) | **85.4** | **76.6** | **83.3** |

**Results of hallucination in VQA.** We further assess AAI on the POPE benchmark across multiple models and decoding strategies. As shown in Table 2, AAI consistently improves average F1 scores, demonstrating strong generalizability. Under the Random, Popular, and Adversarial settings, AAI outperforms both vanilla models and state-of-the-art methods, achieving notable gains.

**Results of Comprehensive Evaluation on MME.** To further assess the model's comprehensive capabilities, we evaluate it on the MME benchmark. As shown in Figure 6, integrating AAI results in notable improvements across both perception and cognition sub-tasks. Specifically, AAI achieves performance gains in certain tasks while maintaining comparable performance in others.

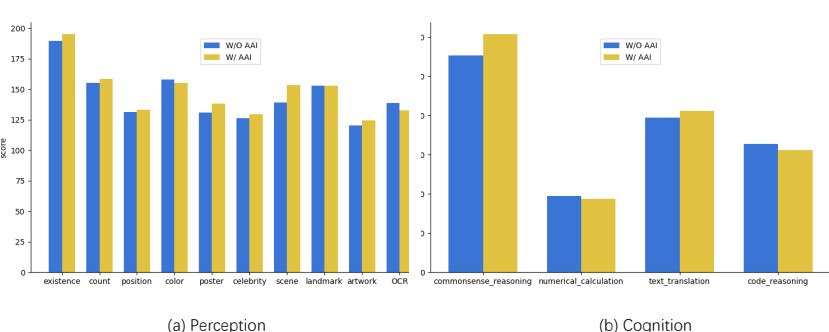

(a) Perception

(b) Cognition

Figure 6: Result on MME.

| Method | LLaVA-1.5 | | | MiniGPT-4 | | | Shikra | | |
|---|---|---|---|---|---|---|---|---|---|
| | A | D | C | A | D | C | A | D | C |
| Vanilla Greedy | 6.08 | **6.22** | 4.54 | 4.86 | 5.14 | 7.12 | 4.82 | 4.52 | 6.38 |
| AAI(Ours) | **6.36** | 6.16 | **5.14** | **5.72** | **5.20** | **7.16** | **5.10** | **5.52** | **6.94** |

Table 3: GPT-4o-assisted hallucination evaluation results on MSCOCO across Accuracy (A), Detailedness (D), and Coherence (C).

**Result of GPT-4o's assistance.** As shown in Table 3, across all models, AAI achieves significant improvements in Accuracy, demonstrating its effectiveness in mitigating hallucinations. For Detailedness and Coherence, AAI yields modest gains in some models and slight declines in certain cases, while still maintaining competitive performance overall. Notably, substantial improvements are observed in Coherence for LLaVA-1.5 and in Detailedness for Shikra. In summary, these results highlight the clear advantages of our approach across all evaluated models.

## 5.4 ABLATION STUDY

**Control Analysis.** To evaluate the effectiveness of AAI, we conduct controlled experiments on LLaVA-1.5 by manipulating its attention alignment. In the first experiment, we remove the image attention used for alignment, while in the second, we perturb the reference mask with Gaussian noise to shift its distribution away from that of the original encoder. We randomly sample 500 images from the MSCOCO dataset and prompt the MLLM

| Method | $CHAIR_S$ | $CHAIR_I$ | $Recall$ |
|---|---|---|---|
| AAI | 37.4 | 9.2 | 78.6 |
| W/O Ref | 44.3 | 11.4 | 73.0 |
| W/ Perturbation | 45.0 | 12.4 | 70.9 |

Table 4: Control Analysis.

to generate captions. As shown in Table 4, both ablation and perturbation lead to clear performance degradation, highlighting the importance of image-guided attention alignment.

**Alignment Coefficient and Reweighting Coefficient.** We evaluate AAI under different values of $\alpha$ and $\beta$ with detailed results provided in the Appendix due to space constraints. Experiments show that increasing $\alpha$ and $\beta$ reduces hallucinations to varying degrees across models, but exceeding certain thresholds compromises generation stability, with the thresholds differing by model. This suggests that we can adjust $\alpha$ and $\beta$ to balance factual accuracy and semantic completeness in different MLLMs.

**Latency and throughput analysis.** To evaluate the efficiency of AAI, we compare its latency and throughput with baseline under different decoding strategies: MemVR with greedy decoding, OPERA with beam search, and VCD with nucleus sampling. Figure 7 presents the results. The overhead introduced by AAI remains well within an acceptable range, striking a favorable balance between effectiveness and cost. Relative to the original decoding, AAI increases latency by approximately 1.1 times, while OPERA and VCD increase latency by about 5.2 times and 1.8 times, respectively. This demonstrates the practical value of AAI in real-world applications.

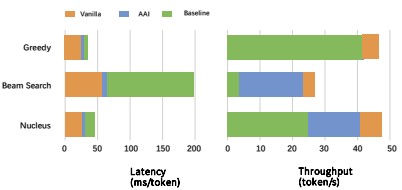

Figure 7: Comparison of latency and throughput across different baselines.

## 6 CONCLUSION

In summary, this paper demonstrates that attention misalignment among image tokens is a key factor contributing to hallucinations in MLLMs, as the model's focus diverges from semantically important visual content. Motivated by this insight, we propose AAI, a training-free method that leverages self-attention from the vision encoder as a reference to guide the model's attention distribution. Extensive experiments validate the effectiveness and generalizability of AAI across diverse MLLMs.

**Reproducibility statement** All experimental code is provided in the Supplementary Materials, with detailed parameter settings available in Section 5 and Appendix C.

**Ethics Statement** This work does not involve human participants or animals. All datasets used in our experiments are publicly available and open-sourced, and no personally identifiable or sensitive information is involved.

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

# A   LIMITATION AND FUTURE DIRECTION

**Limited Experimental Scope.**   Due to computational constraints, our experiments were conducted on a limited set of Multimodal Large Language Models (MLLMs). We did not extend our evaluation of AAI to a broader range of architectures or larger-scale models. We encourage future research to apply and validate our approach across a wider variety of MLLMs to assess its generalizability and scalability.

**Potential for Enhanced Generalization.**   In its current form, AAI leverages the self-attention maps from the vision encoder as guidance signals. While effective, this strategy may introduce bias and limits the method's applicability to multimodal settings beyond vision-language. Inspired by prior work, future research could explore more method to obtain the guidance signals, such as using a lightweight, trainable encoder to generate a more flexible and modality-agnostic reference attention. We aim to extend AAI toward a more unified framework that accommodates diverse input modalities.

**Trade-offs Between Truthfulness and Diversity.**   Despite outperforming existing methods in terms of factual accuracy, our approach exhibits a slight reduction in output diversity and visual detail compared to the original model. This highlights a fundamental trade-off between truthfulness and generative richness. Our future work will focus on mitigating this limitation and achieving a more balanced optimization of both objectives.

# B   COMPARE AND FURTHER ANALYSIS

## B.1   COMPARISON WITH PRIOR STUDY

In the context of LLMs, prior studies suggest that allocating excessive attention to a few 'anchor' tokens can lead to hallucinations in uni-modal language models. Existing work largely argues that MLLMs are progressively influenced by textual priors during decoding, a factor often positively correlated with hallucination. (Zou et al., 2025) To mitigate this, various approaches have been proposed to strengthen or preserve visual information; for example, PAI directly regulates attention to allocate more focus to images.(Liu et al., 2025) Inspired by PAI and related research on LLMs, our study differs in that it focuses on attention distribution within the visual modality rather than across modalities. Consequently, our method can be combined with prior approaches to achieve complementary improvements.

## B.2   FURTHER ANALYSIS

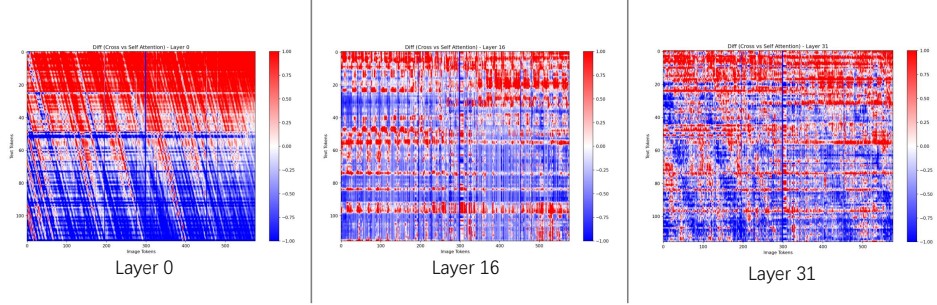

Figure 8: Difference between decoding attention among visual tokens and self-attention in Vision Encoder

Our experiments reveal that the attention shift described in this work does not occur in early layers but gradually emerges in the middle layers. As shown in Figure 8, attention in the early layers remains relatively aligned with the encoder's self-attention, while misalignment arises in the middle layers and becomes pronounced in the final stages. This observation is partly consistent with prior

studies, such as OPERA (Huang et al., 2024) and DeCo (Wang et al., 2024a), which argue that MLLMs exhibit stronger perceptual capability in the early layers than in the later ones. This finding highlights the significance of further investigation in this direction.

# C  DETAILS OF EXPERIMENT

## C.1  EXPERIMENTAL SETTING

**Analytical Experiments.**  For our preliminary analyses, including those presented in section 1,3, we employed LLaVA-1.5, MiniGPT-4, Shikra as the base MLLM and conducted experiments under a greedy decoding strategy. We randomly sampled 500 images from the MSCOCO 2014 dataset for in-depth analysis. For Fig 1(a) and Fig 3, a consistent phenomenon was observed in over 90% of the samples, from which one representative case was selected for illustration. For others, we report results based on the average across all samples.

**Evaluation Experiments.**  For the main evaluation, we benchmarked AAI on CHAIR and POPE using LLaVA-1.5, minigpt4 and Shikra. Our method is applied starting from the second layer of the model. The decoding configurations were as follows:

For **AAI**, we set `do_sample=True`, `max_new_tokens=512`, `use_cache=True`, and either `num_beams=1` (greedy or nucleus decoding) or `num_beams=5` (beam search).

For **MemVR**, we adopted greedy decoding with `do_sample=False`, `temperature=0`, `threshold=0.75`, and `num_beams=1`, following the original implementation.

For **CGD**, we adopted greedy decoding with `do_sample=False`, `temperature=0`, $\alpha = 0.99$, `N=2`, `M=3`, and `num_beams=1`, following the original implementation.

For **HALC**, we adopted greedy decoding with `do_sample=False`, `temperature=0`, $\alpha = 0.05$, `m =0.05`, $\lambda$=`0.6`, `n=4`, and `num_beams=1`, following the original implementation.

For **VCD**, we used the official configuration: `do_sample=True`, `temperature=1`, `noise_step=500`, with the plausibility constraint coefficient set to $\lambda = 0.1$ and the contrastive emphasis to $\alpha = 1$, consistent with the original paper.

For **OPERA**, we followed the settings from the original work, using `beam=5`, `do_sample=True`, `scale_factor=50`, `threshold=15`, `num_attn_candidates=5`, and `penalty_weights=1`.

**Hardware Environment.**  All experiments were conducted on a single NVIDIA RTX 4090 GPU with CUDA 12.1.

## C.2  ABLATION STUDY IN HYPERPARAMETERS

Our method involves two hyperparameters, $\alpha$ and $\beta$. We conducted ablation studies on LLaVA-1.5 under greedy decoding, with results for $\alpha$ and $\beta$ reported in Table 5 and Table 6, respectively. $C_S$ denotes $CHAIR_S$ and $C_I$ denotes $CHAIR_I$.

| $\alpha$ | $\beta$ | LLaVA-1.5 | | | MiniGPT-4 | | | Shikra | | |
|---|---|---|---|---|---|---|---|---|---|---|
| | | $C_S$ | $C_I$ | F1 | $C_S$ | $C_I$ | F1 | $C_S$ | $C_I$ | F1 |
| 0.1 | 1.0 | 46.2 | 12.0 | 78.4 | 29.8 | 9.9 | 70.8 | 55.8 | 14.0 | 75.2 |
| 0.2 | 1.0 | 45.8 | 11.8 | 78.1 | 26.8 | 9.3 | 71.2 | 57.0 | 13.9 | 76.6 |
| 0.3 | 1.0 | 45.1 | 11.8 | 78.0 | 23.8 | 12.0 | 70.7 | 39.8 | 11.7 | 70.2 |
| 0.4 | 1.0 | 40.2 | 10.4 | 76.9 | 21.4 | 9.5 | 70.7 | 10.8 | 8.6 | 39.0 |
| 0.5 | 1.0 | 19.8 | 7.0 | 72.9 | 19.0 | 9.4 | 69.2 | - | - | - |
| 0.6 | 1.0 | 7.0 | 3.4 | 52.9 | 14.5 | 8.3 | 66.1 | - | - | - |

Table 5: Ablation Study of the Hyperparameter $\alpha$.

| $\alpha$ | $\beta$ | LLaVA-1.5 | | | $\alpha$ | $\beta$ | MiniGPT-4 | | | $\alpha$ | $\beta$ | Shikra | | |
|---|---|---|---|---|---|---|---|---|---|---|---|---|---|---|
| | | $C_S$ | $C_I$ | F1 | | | $C_S$ | $C_I$ | F1 | | | $C_S$ | $C_I$ | F1 |
| 0.4 | 0.9 | 18.6 | 4.1 | 71.7 | 0.4 | 0.9 | 25.8 | 9.4 | 70.1 | 0.3 | 0.9 | 25.2 | 8.3 | 68.9 |
| 0.4 | 1.0 | 40.2 | 10.4 | 76.9 | 0.4 | 1.0 | 21.4 | 9.5 | 70.7 | 0.3 | 1.0 | 39.8 | 11.7 | 70.2 |
| 0.4 | 1.1 | 48.0 | 12.1 | 76.9 | 0.4 | 1.1 | 23.8 | 8.2 | 70.8 | 0.3 | 1.1 | 49.6 | 13.2 | 75.3 |

Table 6: Ablation Study of the Hyperparameter $\beta$.

### C.3 WARM UP STEP.

We further conduct an ablation study on the number of
warm-up steps, as shown in Table 7. Incorporating a
warm-up phase leads to improved performance compared
to its absence. However, excessive warm-up steps ad-
versely affect performance, likely due to biased attention
distribution toward the image during the warm-up phase,
which leads to the accumulation of hallucinations in sub-
sequent decoding.

| Warm up Step | $CHAIR_S$ | $CHAIR_I$ | Recall |
|---|---|---|---|
| 0(W/O) | 40.2 | 10.4 | 73.8 |
| 5 | 37.4 | 9.2 | 74.6 |
| 10 | 42.6 | 11.7 | 72.3 |

Table 7: Warm up Step.

### C.4 PROMPT FOR GPT-4O ASSISTED EVALUATION

We introduce the prompt used in the experiments in Table 8 and Table 9

## D CASE STUDY

This Case Study aims to demonstrate performances of AAI intuitively.

## E LLM USAGE

Large language models (LLMs) was employed to check word usage and grammar to ensure clarity
of expression. We acknowledge its contribution in this regard.

No LLMs were used in developing the ideas, research methods, or experimental design of this study.
All research, experiments, and analyses were conducted by the authors, who take full responsibility
for the content of the paper.

Instruction: Please describe this image in detail.

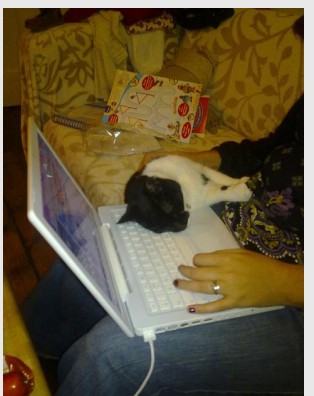

LLaVA-1.5 (Greedy):

The image features a person sitting on a couch with a black and white cat lying on their lap. The cat is resting its head on the person's keyboard, which is placed on their lap. The person is using a laptop, and there is a cell phone nearby.

In the background, there is a chair and a book, possibly indicating a cozy and comfortable environment for the person and their cat.

LLaVA-1.5 (Ours):

The image features a person sitting on a couch with a black and white cat lying on their lap. The cat is resting its head on the laptop keyboard, appearing to be sleeping or relaxing. The person is using a laptop computer, with the cat comfortably positioned on their lap.

In the scene, there is also a book on the couch, possibly a children's book, which adds to the cozy atmosphere of the scene. The person and the cat seem to be enjoying a relaxing moment together.

Input Image

- - - - - - - - - - - - - - - - - - - - - - - - - - - - - - - - - - - - - - - -

LLaVA-1.5 (Greedy):

The image features a snowy landscape with a man standing on a snowboard, likely preparing to ride down the slope. He is wearing a blue jacket and appears to be enjoying the winter sport. There are several other people in the scene, some of them standing in the snow and others walking around.

In addition to the snowboarder, there are two snowboards visible in the image, one near the center and the other towards the right side. A backpack can also be seen in the scene, placed on the ground near the center. The overall atmosphere of the image is that of a fun and active winter day spent outdoors.

LLaVA-1.5 (Ours):

The image features a snowy landscape with a man standing on a snowboard, likely preparing to ride down the slope. He is wearing a helmet, which suggests that he is taking safety precautions while enjoying the activity.

There are also a few other people in the scene, with one person standing further back and two others closer to the man on the snowboard. The group appears to be enjoying their time on the snowy hill, possibly engaging in winter sports or simply taking in the beautiful scenery.

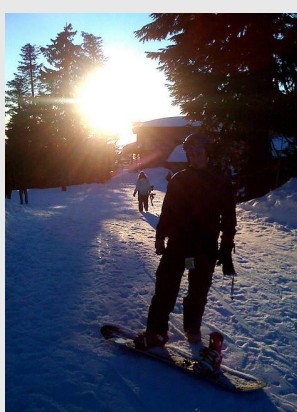

Input Image

Figure 9: AAI's performance in reducing hallucinations of LLaVA-1.5-7B.

Instruction: Please describe this image in detail.

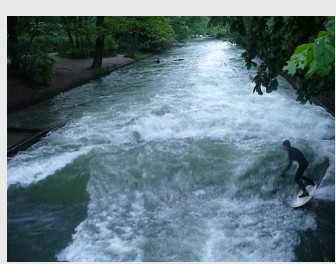

Input Image

Minigpt4 (Greedy):

The image shows a man riding a surfboard down a wavy river. The river is lined with trees on both sides and there are people watching from the shore. The water is turbulent and the man is riding the waves with ease. The sky is overcast and there are clouds in the background. The man is wearing a black wetsuit and a black hat. He has a black beard and is holding onto the surfboard with one hand. The other hand is in the air, as if he is waving to the people on the shore. The river is surrounded by lush green trees and there are some birds flying overhead. The water is a deep blue color and the waves are white and crashing against the shore.

Minigpt4 (Ours):

The image shows a man riding a surfboard down a wavy river. The river is lined with trees on both sides and there are people watching him from the shore. The water is turbulent and the man is wearing a black wetsuit and a black helmet. The sky is overcast and there are clouds in the distance.

- - - - - - - - - - - - - - - - - - - - - - - - - - - - - - - - - - - - - - - - - - - - - - - - -

Minigpt4 (Greedy):

The image shows a group of people standing on the platform of a train station. The train is parked on the tracks in front of them. There are several people standing on the platform, some of whom are carrying bags or packages. The train appears to be a long distance passenger train, with several cars and a locomotive at the front. The people on the platform are dressed in a variety of clothing, including some who are wearing traditional clothing. The scene is set in a rural or small town setting, with buildings and other structures visible in the background.

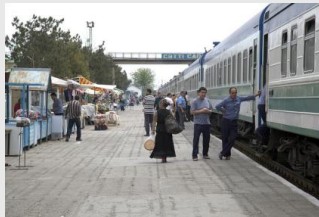

Input Image

Minigpt4 (Ours):

The image shows a group of people standing on the sidewalk next to a train. The train is parked on the tracks and has a green and blue color scheme. The people are dressed in different clothing and are carrying bags and other items. There are also several vendors selling goods in the background.

Figure 10: AAI's performance in reducing hallucinations of Minigpt4-7B.

Instruction: Please describe this image in detail.

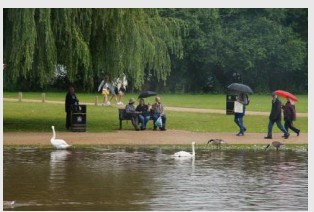

Input Image

Shikra (Greedy):

The image depicts a group of people sitting and walking on a sidewalk near a body of water, possibly a lake or a pond. There are several individuals with umbrellas, indicating that it might be raining or there is a chance of rain. The people are dispersed throughout the scene, with some standing near the water's edge, while others are situated closer to the benches.

In the scene, there are multiple benches and a handbag placed on the ground, providing a place for the people to sit and relax. A bird can be seen swimming in the water, adding a touch of nature to the atmosphere. Additionally, a bicycle is parked nearby, suggesting that someone might have used it to reach the park or to enjoy the view.

Shikra (Ours):

The image depicts a group of people sitting on a bench in a park, near the water. They are enjoying their time by the water, with swans swimming nearby. Each person is holding an umbrella, possibly to shield themselves from the sun or rain. There are a total of six umbrellas, and they are held by at least six different individuals.

In addition to the people sitting on the bench, there are three birds in the scene, with two of them being swans. A handbag can be seen on the ground near one of the people sitting on the bench. The overall atmosphere of the image is leisurely and relaxed.

- - - - - - - - - - - - - - - - - - - - - - - - - - - - - - - - - - - - - - - -

Shikra (Greedy):

The image features a small dog, possibly a Dalmatian, leaping up into the air to catch a yellow frisbee in its mouth. The dog is standing on the grassy area of a playground, with a fence nearby. There are several other frisbees visible in the scene, indicating that the dog is playing in a playful environment.

In addition to the main dog, there are a few other people scattered around the playground, likely enjoying the day or watching the activities.

Shikra (Ours):

The image features a small dog, possibly a Dalmatian, leaping up into the air to catch a yellow frisbee in its mouth. The dog is standing on its hind legs, displaying an impressive jump. In the background, there is a playground area with a slide, adding to the outdoor, playful atmosphere.

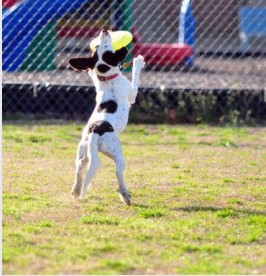

Input Image

Figure 11: AAI's performance in reducing hallucinations of Shikra-7B.

---

**GPT-4o Prompt**

You are required to score the performance of two AI assistants in describing a given image. You should pay extra attention to the hallucination, which refers to the part of descriptions that are inconsistent with the image content, such as claiming the existence of something not present in the image or describing incorrectly in terms of the counts, positions, or colors of objects in the image. Please rate the responses of the assistants on a scale of 1 to 10, where a higher score indicates better performance, according to the following criteria:

1: Accuracy: whether the response is accurate with respect to the image content. Responses with fewer hallucinations should be given higher scores.

2: Detailedness: whether the response is rich in necessary details. Note that hallucinated descriptions should not count as necessary details.

Please output the scores for each criterion, containing only two values indicating the scores for Assistant 1 and 2, respectively. The two scores are separated by a space. Following the scores, please provide an explanation of your evaluation, avoiding any potential bias and ensuring that the order in which the responses were presented does not affect your judgment.

[Assistant 1]
{Response of Assistant 1}
[End of Assistant 1]

[Assistant 2]
{Response of Assistant 2}
[End of Assistant 2]

Output format:
Accuracy: <Scores of the two answers >
Reason:

Detailedness: <Scores of the two answers >
Reason:

---

Table 8: The prompt used for GPT-4o evaluation following Huang et al. (2024); Wang et al. (2024a)

---

**GPT-4o Prompt**

You are required to score the coherence of two AI assistants in describing a given image. Please rate the responses of the assistants on a scale of 1 to 10, where a higher score indicates better coherence.

[Assistant 1]
{Response of Assistant 1}
[End of Assistant 1]

[Assistant 2]
{Response of Assistant 2}
[End of Assistant 2]

Output format:
Coherence: <Scores of the two answers >
Reason:

---

Table 9: The prompt used for GPT-4o evaluation adopted from Wang et al. (2024a)

