# OpenReview forum: "Visual Tokens Are Not Equal: Alleviating Hallucination in Multimodal Large Language Models via Aligning attention"
_ICLR.cc/2026/Conference — ICLR 2026 Conference Withdrawn Submission_

### Official Review · Reviewer_Ez7c · 2025-10-19

**Soundness:** 3
**Presentation:** 2
**Contribution:** 2
**Rating:** 2
**Confidence:** 5

**Summary:**

This paper investigates the phenomenon of multimodal hallucinations, in which the decoder's attention to visual tokens gradually deviates from the visual encoder's self-attention distribution (referred to as "attention misalignment"). The authors propose a training-free decoding method, Align Attention with Image (AAI), which dynamically guides the decoder's attention distribution to visual tokens by caching the visual encoder's self-attention as a reference signal, thereby mitigating the hallucination.

**Strengths:**

Through detailed attention visualization and quantitative analysis (see Figures 1, 3, and 8), the authors reveal a novel mechanism of "attention misalignment," transcending previous perspectives that focused solely on inter-modal imbalance (e.g., text dominance) and providing a new explanatory path for understanding the MLLM hallucination.

This AI approach requires no additional training and only introduces lightweight attention reweighting during inference. It is simple to implement and has low computational overhead (only approximately 1.1x latency), significantly outperforming complex methods such as OPERA (5.2x latency), demonstrating its strong practical value.

**Weaknesses:**

1.The baseline LVLMs used in this paper (LLaVA-1.5, MiniGPT-4, Shikra) are relatively outdated. It is recommended that the authors evaluate the generalization and effectiveness of AAI on more recent baseline LVLMs, such as QwenVL2.5/3 and chatgpt-oss. In addition, the authors are encouraged to further validate the proposed method on video-based MLLMs, such as Video-LLaVA and Video-LLaMA2.

2.The statement “Despite extensive research from various perspectives, the underlying causes remain unclear.” in the abstract is not sufficiently rigorous. The authors should provide a clearer articulation of the research motivation and the limitations of previous approaches.

3.The novelty of this work is limited. Applying attention intervention during the decoding stage to enhance the model’s focus on visual information has already been extensively explored (e.g., [PAI, ECCV], [EAH, EMNLP], [TAME, ICLR], [SEE WHAT YOU ARE TOLD, ICLR], [MemVR, ICML], etc.). The authors should clearly identify the limitations of previous attention intervention methods to better motivate their research.

4.The proposed method is built upon a key observation: during decoding, attention over image tokens shifts away from the encoder’s self-attention, causing the model to overlook critical tokens. However, this conclusion lacks theoretical derivation and interpretability analysis. How are critical tokens defined? The cross-attention between image and non-image tokens during decoding should be further discussed. Moreover, the uneven attention distribution over image tokens is closely related to factors such as RoPE’s long-range decay ([CCA, NeurIPS], [MCA, ACM MM], [Farsight, CVPR]), massive value effects ([Massive Values in Self-Attention Modules are the Key to Contextual Knowledge Understanding, ICML]), and attention sink phenomena ([Label Words are Anchor, EMNLP], [TAME, ICLR]). The authors are encouraged to analyze and discuss these aspects to enhance the theoretical depth of the paper.

5.Many of the conclusions and statements in this paper lack rigorous derivation or empirical validation and appear overly subjective. For example, the claim in the section that “if the model attends to incorrect elite tokens, unintended hallucinations may arise” lacks correlation analysis and a clear definition of incorrect elite tokens. The attention analysis over image tokens should also account for variations across different attention layers and heads. It is recommended that the authors refer to [FastV, ECCV], [MemVR, ICML], and [EAH, EMNLP] for guidance. Furthermore, the claim that a more consistent attention distribution with the encoder leads to fewer hallucinations requires stronger experimental support—the setup in Figure 1(b) is overly simplistic and insufficient to substantiate this conclusion.

6.In Figure 1(a), the legend is labeled as text tokens, yet the description refers to image tokens—this inconsistency needs clarification. What do the different colors represent? In Figure 1(b), what exactly are the hallucinated image tokens and the all image tokens? Are they in a subset relationship? Additionally, the details of the cosine similarity computation should be provided. At present, the experimental setup appears insufficiently rigorous, and the current results do not convincingly support the claimed conclusions. The figure descriptions are also confusing and require clearer explanation.

7.The experiments should include efficiency analyses such as FLOPs and FPS, as well as interpretability analyses including heatmaps and case studies. Comparisons on POPE, CHAIR, and MME should involve more recent hallucination mitigation methods. The generalization ability should be further validated across LVLMs of different scales, and additional experimental results on general benchmarks (e.g., GQA, MMStar, SEED) should be provided.

**Questions:**

see weakness

---

### Official Review · Reviewer_kDEN · 2025-10-31

**Soundness:** 2
**Presentation:** 2
**Contribution:** 3
**Rating:** 4
**Confidence:** 3

**Summary:**

This paper proposes a new method to mitigate hallucination phenomena in MLLMs. The authors argue that a shift between the visual attention distribution of the vision encoder and the visual token attention of the decoder can be a major cause of hallucination.
To address this, they cache the self-attention from the vision encoder and realign the decoder’s attention to match it. Experiments conducted on LLaVA, MiniGPT-4, and Shikra demonstrate that the proposed method effectively reduces hallucinations.

**Strengths:**

This study proposes a method to address the hallucination problem in MLLMs. In particular, it is significant in that it identifies the shift between the visual attention of the vision encoder and the visual token attention of the decoder as a cause of hallucination. Based on this observation, this paper proposes a method that aligns the decoder’s attention with that of the vision encoder, demonstrating a clear reduction in hallucinations compared to the baseline.

**Weaknesses:**

Although Figure 3 is a very important illustration showing the attention shift between the vision encoder and the decoding process, its intended meaning is not clearly conveyed.
- For example, in the hallucinated image part, what exactly indicates that the attention has shifted?
Moreover, compared to the non-hallucinated image, the hallucinated example contains many objects, which makes it confusing to determine what we should focus on in the vision encoder and decoder attention maps.


- In the Method section, the authors mention that the decoder’s attention drifts as the textual context accumulates. How can this be verified? Supporting evidence or analysis seems necessary.


- In the Experiments section, evaluations on more recent models are needed. It would be helpful to test whether the same phenomenon occurs and whether the proposed AAI method can be applied to newer MLLMs such as Qwen2.5-VL and InternVL3.

**Questions:**

1. In Figure 5, why does the number of hallucinated images decrease as the token index increases? Does this phenomenon occur because the number of tokens generated varies for each image?
2. How were the hyperparameters searched? Were they applied directly to the test set? According to the appendix, there appears to be a trade-off between Chair and F1 performance. How were the optimal hyperparameters determined?
3. In the control analysis of the ablation study, is it possible to perturb the image directly instead of perturbing the reference mask?
4. From an efficiency perspective, how does memory usage compare to other methods? Since the AAI method seems to require direct access to attention, is it compatible with FlashAttention? If not, a comparison of memory usage with other methods that support FlashAttention would be necessary.

---

### Official Review · Reviewer_wzU8 · 2025-11-01

**Soundness:** 1
**Presentation:** 3
**Contribution:** 2
**Rating:** 4
**Confidence:** 4

**Summary:**

The paper investigates the problem of hallucination in Multimodal Large Language Models (MLLMs). The authors identify a phenomenon they call "attention misalignment," where the decoder's attention distribution over visual tokens progressively diverges from the vision encoder's own self-attention map. The paper hypothesizes that this shift is a primary cause of hallucinations, as the decoder loses focus on salient visual information.
To address this, the paper proposes "Align Attention with Image" (AAI), a training-free decoding-time method. AAI works by caching the vision encoder's final-layer self-attention map and using it as a "reference signal." During decoding, it modifies the decoder's attention scores over visual tokens to "pull" them back into alignment with this cached encoder map. The method also includes a re-weighting of attention and a "warm-up" phase. The authors claim this method significantly reduces hallucinations on CHAIR and POPE benchmarks without harming general capabilities.

**Strengths:**

- The paper is the first to systematically measure the divergence between encoder self-attention and decoder cross-attention as a correlate for hallucinations.
- The proposed AAI method is training-free and can be applied during inference, which is a significant practical advantage.
- The method (caching one attention map and applying a mask) appears computationally lightweight and efficient.

**Weaknesses:**

1. The method is built on the incorrect assumption that the decoder's attention should mirror the encoder's. This is fundamentally wrong, as the decoder's attention must be dynamic and task-specific, often focusing on non-salient regions that the encoder's global map would ignore.
2. The method claims to use a 5-step "warm-up" (activating at token 6) but also claims to improve performance on the 1-token POPE benchmark. This is a direct contradiction and suggests a severe flaw in the evaluation.
3. By forcing alignment with a global saliency map, the method will actively penalize the model for looking at non-salient objects, even when the prompt requires it. This will likely cause omission hallucinations, a failure mode the paper does not test for or acknowledge.
4. The paper's entire motivation rests on the unproven assumption that misalignment is the cause of hallucinations, rather than a symptom of them.

**Questions:**

1. Please address the central contradiction in the experiment. How can AAI, which is explicitly stated to have a 5-step warm-up, possibly improve results on the POPE benchmark, which requires a single-token ("Yes"/"No") answer?
2. Please justify the core premise. Why should a decoder's task-specific attention (e.g., "find the small object in the corner") be forced to align with an encoder's global saliency map (which will ignore that object)?
3. Did the authors test for omission hallucinations of non-salient objects? Does AAI not, by design, cause this type of hallucination by suppressing attention to regions the encoder found unimportant?
4. How does the paper prove that misalignment is the cause of hallucinations, and not just a symptom of a model that is already hallucinating for other reasons (e.g., language priors)?

---

### Official Review · Reviewer_Mxnp · 2025-11-01

**Soundness:** 3
**Presentation:** 1
**Contribution:** 2
**Rating:** 4
**Confidence:** 4

**Summary:**

This paper identifies the cause of hallucination in multimodal large language models (MLLMs). The authors observe that, during decoding, the decoder’s visual attention gradually diverges from the encoder’s self-attention, leading the model to overlook semantically important image regions and produce hallucinations. To address this, they introduce Align Attention with Image (AAI) — a training-free, decoding-time approach that caches the encoder’s visual self-attention and reuses it to guide the decoder’s attention alignment toward the image. AAI can be combined with various decoding strategies (e.g., greedy, beam search) and consistently reduces hallucinations across benchmarks such as CHAIR, POPE, and MME, without sacrificing generation quality.

**Strengths:**

* The paper presents a clear and well-motivated problem statement, supported by both quantitative analysis and intuitive examples. In particular, the authors convincingly show the difference between hallucinated and non-hallucinated images in terms of alignment between decoder attention and image encoder attention, providing quantitative evidence for their motivation.

* Based on this motivation, the paper proposes an effective method for mitigating hallucination, which is both conceptually simple and empirically validated.

* Experimental results on CHAIR and POPE benchmarks demonstrate that the proposed approach successfully alleviates hallucinations without sacrificing model performance, supporting the method’s efficacy.

**Weaknesses:**

* The presentation quality of the paper is low, which significantly affects readability and perceived polish:
   - (a) Figures are difficult to read, especially due to extremely small fonts that make the contents hard to interpret.
   - (b) There are many inconsistent references throughout the text (e.g., L75 – Fig. 1(a), L145 – Fig. 2, L185 – Fig. 1(b), L212 – Eq. 1, L270 – Eq. 2, L273 – Eq. 3, L282 – Figure 5).
   - (c) Numerous typos remain uncorrected.
   - (d) Table captions are inconsistently formatted (e.g., Table 1 vs. Table 2).

These issues could be easily fixed through proofreading but collectively lower the overall presentation quality.

* Although the proposed method is not specifically designed for object-level hallucination, the main experiments focus only on object-centric benchmarks such as CHAIR and POPE, both emphasizing salient object perception. It would strengthen the paper to include evaluations on attribute, relationship, location, or counter-commonsense hallucination benchmarks (e.g., PhD (Liu et al., 2025)).

* The generalization ability of the proposed method is evaluated only on MME, which is insufficient as a single benchmark. Additional evaluations on at least one or two more benchmarks would better demonstrate robustness across diverse datasets.

* The method introduces more than three hyperparameters, requiring non-trivial tuning. This may reduce its reproducibility and scalability across models.

* Minor: Section 3 mentions LLaVA-1.5, MiniGPT-4, and Shikra without proper citations.

**Reference**:

Liu, Jiazhen, et al. "PhD: A ChatGPT-Prompted Visual Hallucination Evaluation Dataset." Proceedings of the Computer Vision and Pattern Recognition Conference. 2025.

**Questions:**

* In Figure 1(a), how was the visualization computed? Was it based on a single cherry-picked sample, or does it represent an averaged trend?

* For Figure 6, which model and decoding strategy were used to produce the displayed results?

---

### Note · Authors · 2025-11-24

**Comment:**

Thanks for the reviewers’ valuable feedback. We fully recognize that the writing and experiments in this manuscript are not yet sufficiently complete. We will revise the paper by adding more comprehensive experiments and clearer theoretical justification.

**Withdrawal Confirmation:**

I have read and agree with the venue's withdrawal policy on behalf of myself and my co-authors.